# Role of Oligodendrocyte Lineage Cells in Multiple System Atrophy

**DOI:** 10.3390/cells12050739

**Published:** 2023-02-25

**Authors:** Jen-Hsiang T. Hsiao, Onur Tanglay, Anne A. Li, Aysha Y. G. Strobbe, Woojin Scott Kim, Glenda M. Halliday, YuHong Fu

**Affiliations:** Brain and Mind Centre, Faculty of Medicine and Health, School of Medical Sciences, The University of Sydney, Sydney, NSW 2050, Australia

**Keywords:** alpha-synuclein, multiple system atrophy, neurodegeneration, oligodendrocyte, oligodendrocyte progenitor cell

## Abstract

Multiple system atrophy (MSA) is a debilitating movement disorder with unknown etiology. Patients present characteristic parkinsonism and/or cerebellar dysfunction in the clinical phase, resulting from progressive deterioration in the nigrostriatal and olivopontocerebellar regions. MSA patients have a prodromal phase subsequent to the insidious onset of neuropathology. Therefore, understanding the early pathological events is important in determining the pathogenesis, which will assist with developing disease-modifying therapy. Although the definite diagnosis of MSA relies on the positive post-mortem finding of oligodendroglial inclusions composed of α-synuclein, only recently has MSA been verified as an oligodendrogliopathy with secondary neuronal degeneration. We review up-to-date knowledge of human oligodendrocyte lineage cells and their association with α-synuclein, and discuss the postulated mechanisms of how oligodendrogliopathy develops, oligodendrocyte progenitor cells as the potential origins of the toxic seeds of α-synuclein, and the possible networks through which oligodendrogliopathy induces neuronal loss. Our insights will shed new light on the research directions for future MSA studies.

## 1. Background

Multiple system atrophy (MSA) is a rare and aggressive neurodegenerative disease. Its symptomology includes parkinsonism, autonomic dysfunction, and ataxia, progressing rapidly during the disease duration, with an average of ten years [1]. Approximately 50% of patients with MSA will need aids to walk by the third year of motor symptoms, suggesting the necessity of developing early disease modification to enhance the quality of life [2]. The typical neuropathological findings in the post-mortem brains of MSA patients are the glial cytoplasmic inclusions (GCI) positive for α-synuclein and the loss of neurons. The brain regions impacted by these neuropathological changes determine the clinical phenotypes of parkinsonism (MSA-P) and/or cerebellar dysfunction (MSA-C) [3,4]. MSA is primarily a sporadic disease, with limited evidence for a genetic basis and risk genes. However, the prevalence of MSA-P and MSA-C appears to vary in global populations, which shows it to be associated with environments in different ethnic backgrounds [5,6,7]. Clinically probable and possible diagnoses of MSA depend on the manifestation of motor symptoms, levodopa response, and imaging changes [8]. Non-motor symptoms and physical signs indicative of autonomic dysfunction and REM-sleep behavior disorder are often observed in the prodromal stage of the disease [9]. However, currently, there are no reliable biomarkers for a definite diagnosis of MSA. Novel peripheral biomarkers to reflect the accumulation and spread of GCIs in the brain will be promising to assist with disease diagnosis and monitoring.

## 2. Clinical Features of MSA

MSA is typically categorized into clinical subtypes depending on the motor phenotypes (Table 1), i.e., MSA-P, MSA-C, and a mixed subtype of MSA composing features of both MSA-P and MSA-C [4,10]. Patients with olivopontocerebellar atrophy display cerebellar symptoms, including limb and gait ataxia, whereas patients with nigrostriatal atrophy exhibit parkinsonism characterized by bradykinesia, hypokinesia, and rigidity [1,10,11]. Due to the significant overlap in clinical presentation with other parkinsonian diseases, such as Parkinson’s disease (PD), the definite clinical diagnosis of MSA remains a challenge. MSA patients generally respond poorly to levodopa treatment, which is in contrast to patients with PD and is currently one criterion to differentiate the two diseases [12].

There are also MSA variants that do not fulfill these well-known categories. As such, the benign variant of MSA-P (Table 1) has a long disease duration of more than 15 years [13], whereas incidental MSA with GCIs limited to the pons and olivary nuclei [14,15] may be a preclinical phase of MSA similar to incidental Lewy body disease. The non-motor variant of MSA has a predominant autonomic failure (Table 1) and no typical motor symptoms [16], indicating the nigrostriatal regions are spared from GCIs. In contrast, the restricted variant of MSA-P has neuropathology strictly in the nigrostriatal regions [17]. Different from all these variants, the dementia variant of MSA (Table 1) has frontotemporal lobar degeneration and GCIs in the nigrostriatal and olivopontocerebellar regions [18]. The oligodendrocyte pathology is the dominant feature of all these subtypes and variants of MSA, highlighting the underpinning mechanisms these glia contribute to disease pathogenesis.

## 3. Neuropathological Features of MSA

Although different subtypes and variants of MSA vary in the anatomical regions affected by synucleinopathy and accompanied copathology, GCI is the general pathological hallmark [1,3,10,19]. GCIs are primarily composed of misfolded α-synuclein, a protein known to be located in the synaptic terminals of neurons [20]. Due to this neuropathological feature, MSA falls under the umbrella of primary synucleinopathy, a collective name including PD and dementia with Lewy bodies (DLB), both characterized by neuronal α-synuclein aggregations [4,20]. As MSA progresses, the pathological spread of GCIs across the broad brain regions and the increased local density of GCIs highly correlate with the severity of the neuronal loss [4]. Besides the salient GCIs in the MSA brain, α-synuclein aggregation is also observed in the oligodendrocyte nuclei (GNI), neuronal cytoplasm (NCI), and neuronal nuclei (NNI) [1,4]. Interestingly, the distribution of NCIs and NNIs throughout the brain in relation to neurodegeneration is much less striking than that of GCIs and GNIs [21]. This suggests oligodendrocytes play a critical role in neurotoxicity. Hence, oligodendrogliopathy is a primary pathological cause of MSA. The underpinning mechanisms of oligodendrocytes in the pathogenesis of MSA remain elusive. It is also unknown why nuclear aggregation of α-synuclein is specific to MSA but much less often observed in PD and DLB. Oligodendrocyte progenitor cells (OPCs) have been observed close to GCIs in the MSA brain; however, little is known about the role of OPCs in MSA.

## 4. The Implication of Oligodendrocyte Cell Lineage in the Pathogenesis of MSA

Despite their presence in a shared microenvironment, insoluble α-synuclein aggregation has not been observed in OPCs, which has led to the conclusion that GCIs only impact and develop in mature oligodendrocytes [3]. However, the density of OPCs appears to be highly correlated with the number of GCIs, suggesting an association between the presence of GCIs and the capacity to generate OPCs [3]. Parallel to the formation of GCIs, oligodendrocytes also undergo significant morphological changes, where cell enlargement can occur up to sixfold, alongside increased numbers of OPCs in affected brain regions [20]. While this may imply that oligodendrocyte dysfunction locally induces the generation of OPCs, the processes underlying the formation of α-synuclein accumulation in mature oligodendrocytes are still not understood. There has been evidence of endogenous sources of α-synuclein in rodent and human oligodendrocyte lineage cells, with the expression level of α-synuclein significantly decreasing during oligodendrocyte maturation [3,22]. Similarly, the mRNA level of SNCA (the encoding gene of α-synuclein) in the adult mouse cortex is 12:5:1 for neurons, OPCs, and oligodendrocytes, compared with 3:1 for neurons and oligodendrocytes in the adult human cortex, according to Barres Lab’s brain transcriptome database [23]. Since the endogenous cellular level of α-synuclein that contributes to its aggregation is lower in oligodendrocytes compared with neurons, the mechanism leading to enriched α-synuclein being the main component of GCIs is mysterious. This may indicate that delayed or impaired maturation of OPCs into myelinating oligodendrocytes plays a role in forming α-synuclein inclusions within GCIs [3,24]. This has also raised the hypothesis that OPCs contain an earlier pathologic species of α-synuclein, which is soluble but potentially more toxic and prone to transmission.

Although there has been limited information on OPCs in human neurodegenerative disorders and there may be questions about the proliferative potential of these brain cells in the aged human brain, the mean age of MSA onset can be as early as 30 years in MSA-P and 40 years in MSA-C [25]. Accounting for the preclinical phase of MSA when GCIs initiate, the pathological events can start even earlier in life, which is younger than most aging-related neurodegenerative disorders. In addition, abundant OPCs have also been identified in human demyelinating diseases, such as multiple sclerosis, which has an average onset between 20 and 40 years, with some late onset in the 50s [26]. The earlier onset of these conditions, at a time when the proliferative potential of OPCs is perhaps greater, may therefore suggest a key role for OPCs in initiating pathology. This perhaps also explains the biological preference of forming GCIs in MSA rather than neuronal inclusions as observed in PD and DLB if more immature OPCs are indeed the origin of pathologic seeds of α-synuclein.

Our earlier study has shown that MSA brains feature demyelination [20]. Alternatively, α-synuclein inclusions may delay and alter the maturation process of OPCs [3,27], impacting remyelination through downregulation of myelin-associated proteins [24,27]. Failure of OPCs to readily proliferate, migrate, or differentiate will consequently impair remyelination [28], which may further enhance the cellular level of α-synuclein at this stage and continue to signal the brain to generate more OPCs and recruit OPCs to the early-formed loci. Indeed, the impairment of OPC maturation has been suggested as a significant underlying mechanism contributing to the pathogenesis of MSA [27,29].

Although synucleinopathy is the predominant pathological hallmark of MSA, it is not the only neuropathological feature observed in MSA. Other pathological findings include inflammation, dysregulation of iron, and deficient neurotrophic support [30]. The role of inflammation and its timing in disease progression as a causative initiating event or a subsequent resultant reaction needs to be better understood within the context of MSA [31]. OPCs have been found to play a role in antigen presentation and are the cytotoxic targets in inflammatory demyelination [32]. Aberrant oligodendroglial-vascular interactions disrupt the blood-brain barrier (BBB), which can also facilitate central nervous system (CNS) inflammation [33]. In addition, oligodendrocytes are responsible for maintaining iron homeostasis and require iron for the myelination of axons [34]. The total concentration of iron is significantly elevated in areas of the brain affected by MSA, highlighting the impairment of oligodendrocyte function in MSA [21]. As there is a scarcity of knowledge on how OPCs are involved in MSA pathogenesis, it is important to review their biological functions and assess aspects relevant to disease conditions.

## 5. Mammalian Oligodendrocyte Lineage Cells

In the CNS, oligodendrocytes are responsible for myelination, which allows signal transmission and provides neurotrophic support to axons [24,27]. During development, OPCs expressing proteoglycan nerve glial antigen (NG2) differentiate into myelinating oligodendrocytes [28]. However, a pool of OPCs, comprising 5–8% of total glial cells, remain undifferentiated in the adult CNS and retain the ability to generate mature oligodendrocytes [35,36,37,38]. Although myelination and remyelination by mature oligodendrocytes are efficient in young adults, the efficiency of myelination declines dramatically with aging [39].

Demyelination can arise as a primary or secondary pathology. While primary demyelination infers the loss of myelin from an intact axon, secondary demyelination results from the initial axonal loss [28]. Demyelination often occurs in the presence of traumatic or antibody-induced lesions but is also a common neuropathological hallmark found in several degenerative diseases impacting broad brain regions [28]. When demyelination occurs, whether during development, injury, or disease, OPCs readily respond by migrating, proliferating, and differentiating into mature oligodendrocytes to support the remyelination of those exposed axons [3,28,35]. It is worth noting that the remyelination of exposed axons is critical to the survival of the axon, as the exposed axons are otherwise more susceptible to damage, even when neuroinflammation is not present [28].

Several recent studies have conducted comprehensive quantitative proteomic analyses comparing OPCs at different ages [39,40,41,42]. Proteins shown to be markedly downregulated in aged OPCs include the non-erythroid alpha chain of spectrin (SPTAN1, involved in actin stabilization), aldehyde dehydrogenase 1 family member A1 (ALDH1A1, known to promote OPC differentiation during CNS remyelination), alkaline phosphatase (ALPL), folate receptor alpha (FOLR1, a gene mutation known to cause myelination deficits), and transcription factor 4 (TCF4, involved in stage-specific regulation of OPC differentiation). In contrast, aged OPCs upregulate enzymes involved in sphingolipid synthesis, proteins and proteases associated with lysosomal functions, and citrullination protein functions in insulating neurons.

Although oligodendrocytes are mature OPCs, OPCs (also known as NG2+ glia) and oligodendrocytes have been considered independent populations of cells due to the additional characteristics and functions OPCs possess [43,44].

### 5.1. Morphology and Subtypes of OPCs

OPCs arise from the ventricular germinal zones within the embryonic neural tubes [45]. They are evenly distributed in the gray and white matter of the developing and adult brains [46]. OPCs are the most proliferative cell type during homeostasis and pathological conditions [36,37,38].

Morphologically, OPCs have small cell bodies that have filopodia and lamellipodia on the multiple-branched processes [47]. The filopodia allow these cells to survey the local environment and guide their migration [48,49,50]. These dynamic filopods also allow OPCs to achieve a state of homeostasis by detecting nearby OPC density, oligodendrocyte viability, and neuronal axon myelination status [49,50]. When they contact neighboring OPCs, the process with the filopodia retracts [49]. Using embryonic human brain tissue samples, Huang et al. (2020) highlighted that this self-repulsive feature allows OPCs to disperse across the brain [51].

OPCs can be categorized based on their location in the brain. The gray matter OPCs tend to have radial processes, and the white matter OPCs have processes that align with the nerve fibers [45,47]. As the location and morphology of OPCs vary, the function of OPC subtypes may also differ. The current literature on the subtypes of OPCs is minimal. Future studies will be needed to examine the roles of different subtypes of OPCs in brain regions with different disease vulnerabilities.

### 5.2. Biological Functions of OPCs

The main role of OPCs is to produce mature myelinating oligodendrocytes throughout one’s lifetime to be able to remyelinate upon injury [44,49]. OPCs form surveillance networks for CNS injuries and tissue repair [49]. Recently, many other functions were revealed to be independent of myelination. For instance, fine-tune neural circuits and axon arbor size perhaps by engulfing and pruning axons [52,53]. OPCs also have an immunomodulatory capacity as they express cytokine receptors and can cross-present antigens to cytotoxic CD8+ T cells [54]. Furthermore, OPCs have synaptic signaling properties due to their connections to neurons. OPCs also synapse with glutamatergic neurons in the gray and white matter. A transgenic mouse model highlighted that the pattern of neuronal activity in the brain region influences glutamate release and the differentiation of OPCs [55]. This may elucidate the differences in the cell cycle of OPCs between white matter and gray matter, with a higher rate of proliferation and differentiation in the white matter compared with the gray matter [46].

Recent studies have revealed that different OPCs respond to neurons differently [56]. OPCs possess heterogeneous functions between brain regions, and the OPC-expressed ion channel density predicts the functional state of the OPC [57]. McKenzie and colleagues found that blocking OPC differentiation into oligodendrocytes impairs motor skill learning in mouse models [58]. Consistent with this, Lewis and colleagues showed an association between OPC differentiation and motor learning [59]. In addition, OPCs in multiple sclerosis patients have impaired differential ability to mature oligodendrocytes [60]. These findings suggest that OPC pathology could be associated with motor symptoms.

In addition, ischemic stroke models in mice have revealed that OPCs play a role in maintaining BBB integrity [61]. Wang and colleagues found that OPC transplantation can be a potential treatment intervention for ischemic stroke. This is because OPCs can activate the Wnt/β-catenin pathway, which increases BBB tight junction proteins to help prevent BBB leakage [61].

### 5.3. Morphology and Subtypes of Oligodendrocytes

Most of the literature categorizes oligodendrocytes based on their maturation stage [62] (Figure 1). Following proliferation, NG2+ OPCs will turn into pre-oligodendrocytes (A2B5+), immature oligodendrocytes (O4+), and, finally, mature myelinating oligodendrocytes that express myelin basic protein (MBP) [62,63,64]. The mature myelinating oligodendrocytes can be categorized differently based on their morphological characteristics, locations in the CNS, and functional differences. 

Based on morphological characteristics, del Río-Hortega classified oligodendrocytes into four types (Figure 1). Small and rounded Types I and II oligodendrocytes compose the main population in the white and gray matter [65]. Type I oligodendrocytes form myelin segments on small-diameter axons in different orientations, whereas Type II oligodendrocytes are exclusively in white matter, forming parallel myelin segments [66,67]. Type III oligodendrocytes have one or more processes that do not usually branch. These cells myelinate fewer large-diameter axons [65,67]. Type IV oligodendrocytes do not have processes and form a single long myelin sheath to myelinate only one axon with a large diameter [67]. Similar to OPCs, the location and function of these subtypes of oligodendrocytes have not been extensively studied. Future studies investigating different subtypes of oligodendrocytes would further elucidate the pathomechanism of oligodendrocyte dysfunction-related diseases.

**Figure 1 cells-12-00739-f001:**
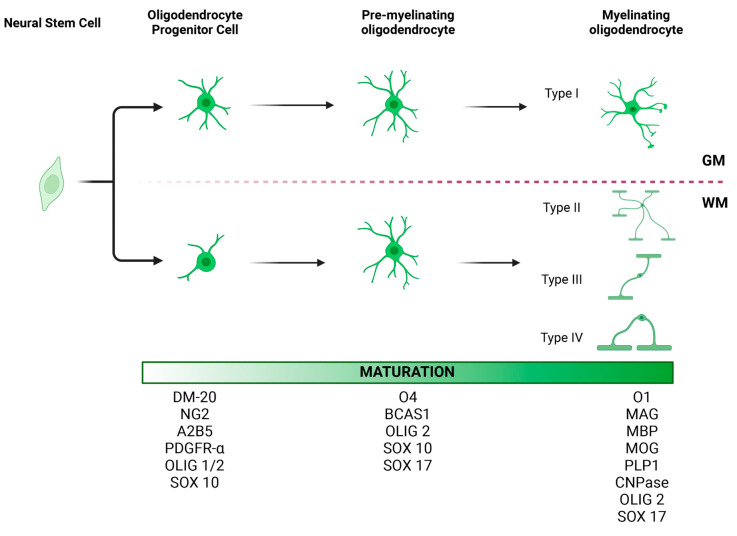
Morphology and maturation of oligodendrocyte lineage cells (OLCs). At least three stages of oligodendrocyte differentiation have been described: oligodendrocyte progenitor cells (OPCs), pre-myelinating oligodendrocytes, and mature myelinating oligodendrocytes. Each stage is characterized by the expression of certain protein markers, which may be useful in identifying these cell types in the brain tissue. OLCs also vary in morphology depending on their location. OPCs in gray matter tend to have radial processes. In contrast, OPCs in white matter tend to have processes parallel to axonal fibers. Accordingly, four different types of oligodendrocytes have been described, with Type I oligodendrocytes being the most dominant in the gray matter and Type II oligodendrocytes in the white matter [67,68]. While the location of Type III and IV oligodendrocytes has not been well characterized, we hypothesize that given their morphology and longitudinal processes, they too are most commonly seen in the white matter. GM, gray matter; WM, white matter. This figure was created with BioRender.com.

Location-based categories include white matter and gray matter oligodendrocytes (Figure 1). Oligodendrocytes are the dominant cell type in the white matter [69]. The lipid-rich myelin produced by oligodendrocytes contributes to the pale appearance of the white matter. The oligodendrocytes in the gray matter are mainly responsible for regulating the neuronal microenvironment rather than myelination [70].

Moreover, oligodendrocytes can be categorized based on their functions and ability to produce myelin. Myelin-forming oligodendrocytes produce myelin sheaths, whereas OPCs and non-myelinating ‘perineuronal’ or ‘satellite’ oligodendrocytes generate myelin-producing oligodendrocytes upon injury, development, or plasticity during one’s lifetime [65].

### 5.4. Oligodendrocyte Functions

The main function of mature white matter oligodendrocytes is myelination, while gray matter oligodendrocytes wrap cells and their processes locally [71,72]. By generating plasma membranes with a large lipid content and wrapping them around the targeted parts of neurons, oligodendrocytes produce a myelin interface for optimal function of the entire neuron [54,71,72]. The myelin sheath is an extended membrane of the oligodendrocytes that provides electrical insulation and thereby allows rapid signal transduction [54]. Oligodendrocytes have also been shown to play a role in metabolic support [73], uptake of fatty acids and lipid metabolism [68], and production of neurotrophic factors [74].

## 6. Oligodendrocyte Lineage Cells in Other Diseases

In healthy individuals, new oligodendrocytes must be produced from the OPC pool in the CNS to remyelinate neurons, prevent axonal degeneration, and preserve normal functions [54]. Upon CNS injury, there will be an increase in local OPC proliferation [45] and recruitment of OPCs to the injury site [37]. The recruited cells do not exhibit self-repulsive characteristics in an attempt to restore normal oligodendrocyte density [49]. When the adjacent OPC is removed due to death or differentiation, the nearby OPC proliferates to replace the lost cell and achieve homeostasis [49]. This proliferation of OPCs occurs in the presence of CNS demyelination, traumatic injuries, and chronic neurodegenerative diseases. This has been demonstrated by in vivo studies in an amyotrophic lateral sclerosis mouse model, which revealed enhanced OPC proliferation and accelerated differentiation to oligodendrocytes at the injury site [75]. Demyelinating diseases, such as multiple sclerosis, schizophrenia, and Alzheimer’s disease, are all closely associated with reduced/dysfunctional OPCs and oligodendrocytes [54]. Chang and colleagues proposed a possible disease cause in multiple sclerosis, highlighting the reduced number of OPCs in lesions [76].

## 7. Methylation Effects on Oligodendrocyte Lineage Cells

The maturation process of OPCs is a prerequisite for remyelination in demyelinating disorders, which is coordinated by complex intracellular transcription factors, extracellular signals, and epigenetic mechanisms [77,78]. MSA is a disease without a renowned genetic preference. OPCs’ proliferation and myelination of axons by mature oligodendrocytes are highly receptive and adaptive to environmental stimuli affecting neuronal activities. Epigenetic changes are known to be very important, through which environmental stimuli are carried out [77,78]. Here, we summarize the potential risk factors revealed by recent methylation studies.

Recent studies have focused on the importance of epigenetic mechanisms and their non-genetic regulation of gene expression at the cellular level [77,79,80]. These studies revealed epigenetic regulators during development and regeneration in response to environmental changes, including covalent modifiers of DNA methylation, histone-modifying enzymes, chromatin modifiers, and non-coding RNA (ncRNA) regulators. However, it remains elusive about the epigenetic mechanism in terms of how environmental changes trigger OPC differentiation and oligodendrocyte myelination.

DNA methylation has been the hub of human epidemiological epigenetic research. The first study linking DNA methylation to oligodendrocyte development was conducted in a neonatal rat model, showing hypomyelination and disrupted oligodendrocyte genesis [81]. In this model, the inhibitor of DNA methyltransferases (DNMT), 5-azacytidine, was applied, suggesting oligodendrocyte lineage cells are vulnerable to DNA methylation. Dnmt1 and Dnmt3a/b are the most distinct forms of DNMTs, which are responsible for maintaining DNA methylation by adding a methyl group to cytosine (5mC). Transgenic mice lacking Dnmt3a showed impaired oligodendrocyte differentiation and dysfunction in remyelination after injury [82]. In contrast, the ablation of Dnmt1 showed altered splicing events such as intron retention and exon skipping in genes involved in myelination, cell cycle, and lipid metabolism, indicating the crucial role of DNA methylation during neonatal oligodendrocyte development [83].

Histone modification includes post-translational changes of histone tails by deacetylation, ubiquitination, phosphorylation, and methylation. Histone deacetylation of the lysine residue is the most prevalent type of histone modification [82,83,84]. The process of acetylation is established by histone acetyltransferases (HATs), whereas deacetylation is maintained by histone deacetylases (HDACs). HDACs have been shown to be involved in oligodendrocyte development, as their inhibition can potentially decrease oligodendrocyte maturation and differentiation [85]. Treatment with HDAC inhibitors in vitro can suppress inhibitory transcription factors, preserving OPCs in a proliferative and undifferentiated state during the early onset of oligodendrocyte lineage progression [86].

A recent study has shown that N6-methyladenosin (m6A) modification on mRNA has regulated OPC differentiation, whereas the deletion of Prrc2a (m6A reader) and Mettl14 (m6A writer) decreased mature oligodendrocytes and induced hypomyelination by regulating oligodendrocyte transcription factor 2 (Olig2) expression in an m6A-dependent manner in vitro and in vivo [87].

## 8. Alpha-Synuclein and Oligodendrogliopathy

The pathological hallmark of MSA includes GCIs [88], which feature α-synuclein aggregates. With little evidence of α-synuclein expression in the human OPCs, it is unclear how the toxic form of α-synuclein initiated in the oligodendrocyte milieu propagates to form insoluble GCIs and then broadly spreads to other brain cells. Previously, knowledge that the endogenous α-synuclein expression is exclusively in neurons has facilitated the neuron-centric dogma that pathologic α-synuclein released from surrounding neurons is taken up by oligodendrocytes as the source of GCIs [22,89]. A recent finding in a mouse model revealed the other potential route for oligodendrocytes to succeed neuronal α-synuclein by pruning α-synuclein-containing neurites [90]. However, there are also proofs supporting GCI’s oligodendrocyte lineage origin. For instance, mature oligodendrocytes normally do not take up extracellular α-synuclein preformed fibrils (PFF) as neurons do. Instead, OPCs can take up PFFs, leading to inclusion formation [24]. These inclusions remain even after the maturation of these OPCs, providing a potential origin of pathologic α-synuclein seeding in MSA [91].

Recent studies have shown that OPCs treated with PFFs had an upsurge and multimerization of their endogenous α-synuclein, which interfered with the expression of proteins associated with neuromodulation and myelination [21,24,92]. PFF-treated OPCs have shown reduced autophagic proteolysis, deficient differentiation efficiency, and newly differentiated mature oligodendrocytes containing α-synuclein accumulation but reduced myelin-associated proteins, such as MBP and tubulin polymerization promoting protein (TPPP/P25) [24,80]. Immunoelectron microscopy confirmed that PFF treatment induced α-synuclein expression on the cell membranes and upregulated endogenous levels in the cytosolic matrix of OPCs but not in mature oligodendrocytes, whereas exposure to monomers did not induce enhanced α-synuclein immunoreactivity in OPCs [24]. Despite the accumulation of α-synuclein in OPCs being confirmed to contribute to the formation of GCIs, immunoblotting and immunostaining showed that these aggregations were not composed of phosphorylated α-synuclein as typically seen in post-mortem brains of MSA patients [24]. This may suggest that both α-synuclein monomers and PFFs are incapable of initiating the phosphorylation. Other pathologic or microenvironmental factors might be required to coexist to propagate the pathologic modification.

## 9. Oligodendrocyte Lineage Cell MSA Models

To date, in vitro and in vivo MSA models mainly focus on investigating the effects of modifying α-synuclein (either wild type or SNCA mutations) by targeted expression in oligodendrocytes and OPCs [22]. The (Plp)-α-Syn transgenic mouse has specific oligodendroglial overexpression of human α-synuclein under the control of the proteolipid promoter. The mouse line has proven useful in studying MSA-related pathogenic mechanisms and is the most extensively characterized preclinical/prodromal MSA model. The representative MSA features in this model include GCI-like pathology, microglial activation, loss of trophic support, nigrostriatal degeneration, a progressive motor phenotype, and early autonomic dysfunction [3,93]. Although overexpression of α-synuclein in oligodendrocytes may be a simplified model that cannot fully mimic the actual pathogenesis of MSA, the presence of α-synuclein aggregates forming GCIs is the critical event in MSA pathology linked to the presented central cardiovascular autonomic failure [22,94]. The model proves that oligodendrogliopathy is sufficient to induce MSA-like prodromal symptoms and that MSA is a primary oligodendrogliopathy disorder.

Induced pluripotent stem cells (iPSCs) from MSA patients have shown great cell modeling potential via the application of four transcription factors: Oct3/4, Sox2, Klf4, and C-Myc. Subsequent differentiation of the iPSCs into OPCs after 60 days was performed on fibroblast cells from MSA familial patients and characterized by immunocytochemistry with OPC markers and the bipolar morphology of these immature cells [95]. The transcript level of SNCA was subsequently downregulated with the maturation of the oligodendrocytes [95,96].

MSA pathogenesis is also known to be associated with mitochondrial dysfunction and a reduction in respiratory chain complex I activity, as shown in the skeletal muscle of MSA patients [97]. Rat models targeting nigrostriatal mitochondria by striatal injection of succinate dehydrogenase inhibitor 3-nitropropionic acid or mitochondrial complex I inhibitor (MPP+) induce extensive neuronal loss in the substantia nigra and striatum as well as a motor deficit resembling MSA-P [98]. The iPSCs-derived neurons from the patient with a variant of *COQ2* MSA have shown mitochondrial dysfunction [99,100]. In this study, Nakamoto and colleagues reprogrammed peripheral blood mononuclear cells into iPSCs, followed by differentiation into different midbrain and hindbrain neurons, including glutamatergic, GABAergic, dopaminergic, and glycinergic neurons. The *COQ2* variant patient-derived neurons were shown to have reduced mitochondrial mass, COQ10, and oxygen consumption rate and presented an extracellular acidification [101]. Unfortunately, the mitochondrial model has not been applied to oligodendrocyte lineage cells; therefore, there is currently missing proof of whether mitochondrial deficiency or dysfunction is one of the underpinning mechanisms of oligodendrogliopathy involved in MSA.

## 10. The Factors Involved in the Process of OPC Maturation

Much of the literature on gliogenesis in MSA comes from murine models. These studies have established at least three stages of oligodendrocyte lineage differentiation (Figure 1). OPCs are characterized by markers such as NG2, A2B5, PDGFRα, Olig1, Olig2, and Sox10. Pre-myelinating oligodendrocytes can be identified using BCAS1, O4, Olig2, Sox10, and Sox17. Mature myelinating oligodendrocytes express O1, MAG, MBP, MOG, PLP1, CNP, Olig2, and Sox17 [7,102]. Indeed, at least 12 clusters of oligodendrocytes have been described through transcriptomic analysis, though the morphology and function of each remain poorly understood [103,104,105,106].

In mice, OPC formation occurs in three waves through asymmetric division of radial glia. Interestingly, Li et al. recently found a common progenitor for astrocytes, oligodendrocytes, and olfactory bulb interneurons [107,108]. The development of the oligodendrocyte lineage has recently been reviewed by Kuhn, Gritti, Crooks, and Dombrowski [54]. Briefly, newborn OPCs express DM-20 mRNA, an isoform of PLP. Commitment to the oligodendrocyte lineage is characterized by the induction of Sox10 by Olig1 and Olig2. Sox10 subsequently induces Cspg4, which encodes NG2. PDGFRα is a receptor for PDGF-A, an OPC survival marker produced by neurons and astrocytes [109,110]. Olig1 and Olig2 are abundant transcription factors throughout the oligodendrocyte lineage. While the role of Olig1 is less known, Olig2 is essential for OPC differentiation and inducing functional gain to promote remyelination in mice [111].

A recent transcriptomic analysis of embryonic human OPCs found that in addition to a cluster of cells expressing the conventional OPC genes OLIG1, OLIG2, PDGFRA, NKX2-2, SOX10, S100B, and APOD, there was an additional cluster of pre-OPC cells expressing OLIG1, OLIG2, PDGFRA, in addition to EGFR [51]. These pre-OPC cells are also expressed in the outer radial glial cells (oRGs), suggesting that oRGs, which are rarely found in rodents and are thought to contribute to gray matter expansion in primates, may provide an additional source of oligodendrocytes in addition to the outer subventricular zone [51]. Other OPC differentiation factors that have been identified include triiodothyronine (T3) [112], adenosine receptor ligands [113,114,115], and AMPA receptors [102].

The involvement of OPCs in MSA has long been suspected [21]. In fact, a recent CSF analysis in 50 patients with MSA revealed higher levels of NG2 and neurofilament-L (NF-L) than controls [116]. While NF-L as a marker of axonal injury is well established, increased detection of NG2 may be due to increased OPCs in MSA. Certainly, increased numbers of OPCs in the brains of MSA patients have been reported [27]. Such evidence of the potentially increased activity of OPCs in MSA, along with the much higher α-synuclein expression in OPCs compared with oligodendrocytes, points to a contribution of OPCs in MSA pathogenesis. This overexpression of α-synuclein has been shown to delay OPC maturation by downregulating the myelin-gene regulatory factor and MBP [27,29]. Interestingly, a recent analysis of O4^+^ pre-myelinating oligodendrocytes generated from patient-derived iPSCs demonstrated that overexpression of α-synuclein induced the upregulation of several genes important for OPC maturation into O4^+^ oligodendrocytes, including MBP, MOG, MAG, CNPase, and NKX2.2, and downregulated astrocyte genes [117]. This suggests that α-synuclein may play a physiological role early in the maturation of OPCs, where increased levels of α-synuclein may be necessary for the commitment of cells to the oligodendrocyte lineage, whereas later lower levels are necessary for their complete maturation into myelinating oligodendrocytes. The same study also demonstrated that instead of maturing into myelinating oligodendrocytes, O4^+^ pre-myelinating oligodendrocytes treated with α-synuclein fibrils transformed into an antigen-presenting phenotype. Early immunoreactivity of OPCs may therefore be crucial in the pathogenesis of MSA. However, this also demonstrates the need to consider the heterogeneity of OPCs in MSA, as it is becoming increasingly recognized in multiple sclerosis [104,118]. Another consideration is the differences in the transcriptomic profile of embryonic and adult OPCs, with Lin et al. demonstrating that neonatal primate O4^+^ cells preferentially expressed genes involved in differentiation and proliferation, whereas adult O4^+^ cells had higher expression of genes involved in cell death and survival [119].

## 11. MSA Risk Genes Associated with Oligodendrocyte Lineage

In order to gain further insight into the oligodendrocyte lineage-related pathways involved in MSA pathogenesis, gene-based pathway prediction was performed. Risk genes were extracted from the literature, DisGeNet [120] and the Harmonizome database [121], and subsequently used Brain RNA-Seq [23] to determine the oligodendrocyte-specific expression of these genes in humans, though data for OPCs were not available. All genes with an FPKM greater than one were used for pathway prediction with QIAGEN IPA (QIAGEN Inc., Hilden Germany, http://digitalinsights.qiagen.com/IPA (accessed on 5 December 2022) [122]). This process yielded seven distinct networks, broadly falling into protein aggregation, DNA methylation, iron homeostasis, neuroinflammation, myelin formation, cell maturation, and glutaminergic excitotoxicity (Table 2 and Figure 2).

Apart from MBP, only a few OPC-associated MSA risk genes we identified have been discussed in the literature. Most recently, Piras and colleagues demonstrated the downregulation of QKI in MSA-C following RNA expression profiling of cerebellar white matter [123]. QKI is known to regulate oligodendrocyte differentiation and myelination, though it may also have a potential role in regulating RNA metabolism in these cells. It may also be a potential therapeutic target in MSA, as Zhou et al. found that PPARβ and RXR agonists were able to alleviate QKI deficiency-induced demyelination in QKI knockout mice [124]. ATXN2 gain-of-function mutations are observed in spinocerebellar ataxia and amyotrophic lateral sclerosis, though they have also been identified in MSA [125]. Functionally, ataxin-2 has been associated with lipid metabolism and may inhibit myelin formation by repressing mTORC1 signaling [126]. Fyn kinase has been implicated as an effector in Aβ-induced remyelination and OPC proliferation and differentiation [127]. While this may be important in Alzheimer’s disease pathology, Aβ can be an observed copathology in MSA cases surviving to older ages.

In addition to these genes, a recent unpublished study performed single-nuclei sequencing of putamen oligodendrocytes in MSA, PD, and controls [128]. Their results demonstrated the downregulation of genes associated with regulating apoptosis and senescence in MSA OPCs, including EGFR, NFKB, STAT3, FOXO1, MDM2, CDKN2A, TNIK, TXNIP, and BCL9. This points to either an impact of α-synuclein or another disease factor dysregulating the OPCs, or the accumulation of α-synuclein may be a consequence of cell senescence. Mature oligodendrocytes show increased expression of proteins associated with neurogenesis, oligodendrocyte differentiation, and myelination, which may reflect compensatory processes to counteract pathology.

## 12. Targeting the Oligodendrocyte Lineage in MSA Treatment

Oligodendrocytes impacted by MSA are well established, but there have been limited attempts to enhance oligodendroglial differentiation and remyelination in MSA. In the context of multiple sclerosis, however, multiple studies have identified molecules that may be able to promote oligodendrocyte differentiation, and there are also several ongoing animal and human trials (recently reviewed by [129]). Some of these include anti-muscarinic agents, such as benztropine and clemastine, which have been shown to increase MBP expression and promote OPC differentiation in rats; and Opicinumab (BIIB033), a blocker of LINGO-1, which is normally expressed by oligodendrocytes as a negative regulator of their differentiation and myelination; and histamine H3 receptor antagonists. So far, none of these agents has demonstrated sufficient efficacy in multiple sclerosis, and none has reached clinical use. Indeed, this reflects the ubiquitous struggle to identify disease-modifying therapies for all neurodegenerative disorders in general. It is likely that this stems from an incomplete understanding of such complex disorders, an attempt to distill the several pathogenic processes associated with them into a single target, and the inability to identify patients amenable to therapy early in the disease course. In the context of MSA, future studies should establish the relationship between the genetic influences implicated in MSA and OPC differentiation in association with α-synuclein aggregation (Figure 2).

Aside from pharmacotherapy, stem cell transplantation may be another therapeutic avenue for MSA. Previously, intra-arterial administration of autologous mesenchymal stem cells demonstrated some benefit in MSA [130], and an evaluation of its intrathecal administration is underway. The therapeutic success of stem cell implantation may, however, be improved by the direct implantation of OPCs, given the advances that have been made in differentiating human pluripotent stem cells into OPCs. While this has shown remyelination potential in mice [61], it has not yet been undertaken in humans and requires further refinement to ensure efficacy.

In conclusion, oligodendrocyte lineage cells are key players in the pathogenesis of MSA. The knowledge of the oligodendrocyte pathways and mechanisms involved in the development of synucleinopathy is extremely limited. Future studies revealing the role of the oligodendrocyte lineage in MSA pathogenesis will provide insights into developing novel pharmacological targets and early biomarkers for the disease.

## Figures and Tables

**Figure 2 cells-12-00739-f002:**
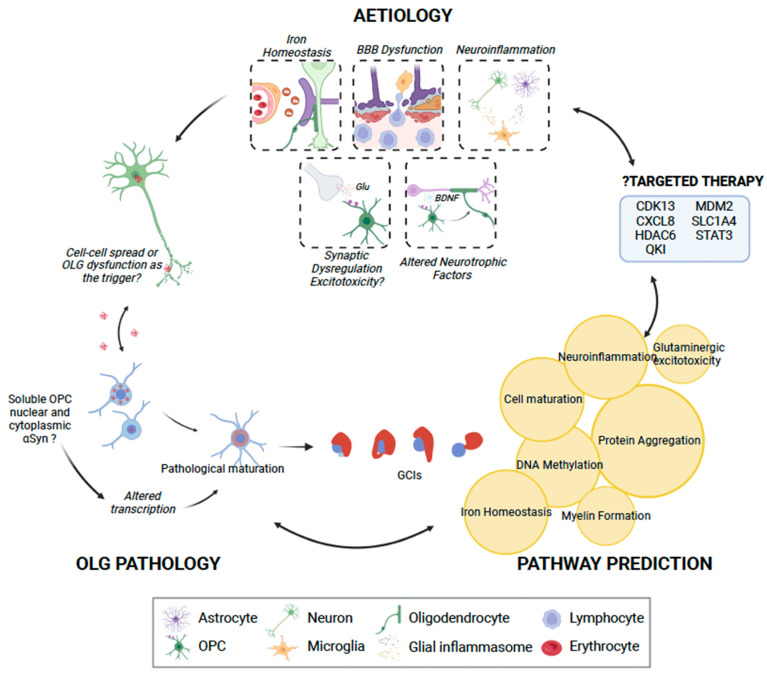
The role of oligodendrocyte lineage cells (OLCs) in MSA pathogenesis. While OPCs may not be able to uptake insoluble α-synuclein, they may play a role in pathogenesis by uptaking and generating toxic soluble α-synuclein. This may in turn result in pathological maturation of oligodendrocytes, with consequent glial cytoplasmic inclusions, or possible direct neuronal spread of α-synuclein. We identified seven key clusters of pathways through IPA [122], which may be disrupted due to OLC dysfunction (highlighted in yellow circles), consequently manifesting in interruption of key physiological processes and, finally, neuronal loss. IPA analysis also identified seven genes that may be possibly targeted using pharmacotherapy. OLG, oligodendrocyte; OPC, oligodendrocyte progenitor cell; Glu, glutamate; BDNF, brain-derived neurotrophic factor. This figure was created with BioRender.com.

**Table 1 cells-12-00739-t001:** Clinical variants of multiple system atrophy.

Clinical Subtype	Main Motor Feature	Pathological Distribution
MSA-P	Parkinsonism	Nigrostriatal atrophy
MSA-C	Limb and gait ataxia	Olivopontocerebellar atrophy
Mixed subtype	Both motor features	Combined atrophy
MSA-autonomic failure	No motor features	Central autonomic brain regions
MSA-dementia	No motor features	Frontotemporal plus combined atrophy

**Table 2 cells-12-00739-t002:** Oligodendrocyte lineage genes with predicted pathways involved in MSA.

MSA Associated Genes	Networks	Pathway Prediction	Genes with Known Target Drugs
ATXN3, CRYAB, HDAC6, MDM2, NEDD8, RAB29, SNCA, STRN, TPPP	Protein Degradation, Cellular Assembly and Organization, Cellular Function and Maintenance,	Protein Aggregation	HDAC6, MDM2
CDK13, CDKN2A, FMR1, MBP, NDUFB11, STAT3	Cell Cycle, Cellular Assembly and Organization, Hematological System, Development and Function	Iron homeostasis, Cell maturation	CDK13, STAT3
CXCL8, QKI, SLC1A4, TXNIP	Cardiovascular System, Development and Function, Organismal Development, Tissue Morphology	Neuroinflammation, Glutaminergic excitotoxicity	CXCL8, QKI, SLC1A4
ATG4C, COQ2, SMUG1	DNA Methylation, Dermatological Diseases and Conditions, Developmental Disorder, Recombination, and Repair	DNA Methylation	
ABCA8	Lipid Metabolism, Molecular Transport, Small Molecule Biochemistry	Myelin formation	

## Data Availability

No new data were created or analyzed in this study. Data sharing is not applicable to this article.

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
