# Peer review of "Role of Oligodendrocyte Lineage Cells in Multiple System Atrophy"

_cells, 2023, doi:10.3390/cells12050739_

Round 1
Reviewer 1 Report
This is an excellent and timely overview of the role of oligodendrocytic lineage celle in the pathogenesis of MSA, an oligodendrogliopathy with secondary neurodegeneration including demyelination. After a short description of the clinical and neuropathological features of MSA, the maturation of human oligo lineage cells, subtypes of oligos, oligodendrocytic function and dysfunction, methylation effects on oligos, the interplay between aSyn and oligos, factors involving their maturation, the few MSA risk factors associated with oligo lineage, oligo-related MSA models, and finally, possible role of oligo lineages in MSA treatment (as well as stem cell therapy) are extensively and critically reviewed. The relevant literature is widely considered, and the different veiws about the role of oligos in the pathogenesis of mSA are critically considered. The present paper is a valuable contribution to our knowledge about the role of oligodendroglia in the hitherto not fully explained pathogenesis of MSA.
Author Response
We thank the reviewer for appreciating our review and hope it will introduce innovative insights to the field.
Reviewer 2 Report
Page 2,
Section 1. " Clinical features of MSA" would benefit from a table.
Page 3:
"However, the density of OPCs appears to be highly correlated with the number of GCIs, suggesting a causative association between the presence of GCIs and the capacity to generate OPCs" --> How does density alone suggest causative association between presence of GCIs and capacity to generate OPCs? Can the authors elaborate a bit more?
" ..... This perhaps also explains the biological preference of forming GCIs in MSA rather than neuronal inclusions as observed in PD and DLB if more immature OPCs are indeed the origin of pathologic seeds of a-synuclein" --> the presence of more or less OPCs does not explain a preference for forming GCIs in MSA instead of neuronal inclusions. Recently, Azevedo and colleagues demonstrated that important cellular pathways and network dysfunction occur in MSA iPSC-derived OPCs which clearly suggests cell autonomous dysfunction as a starting point for the build-up of aggregates. Interestingly, the cells adopted an antigen presenting phenotype due to the aggregates formation and were delayed in their maturation, in line with work from Ettle, Winkler and colleagues.
Page 6
figure 1 legend. please correct typo: OLGs should be replaced by OLCs
Line 328: change inhibitor for inhibition
Line 413: remove extra space
Line 414-415: This tripotential basal multipotent progenitor cell may provide further insights into the anosmia associated with a-synucleinopathies. --> please explain this sentence. what is the relation between anosmia and the formation of OPCs and tripotent stem cells? alternatively, please delete the whole paragraph except the first sentence.
Author Response
Reviewer comments
Page 2, Section 1. " Clinical features of MSA" would benefit from a table.
Authors' note to Reviewer 2: we thank the reviewer for this suggestion. We have added the Table below at the end of the clinical section.
Clinical subtype |
Main motor feature |
Pathological distribution |
MSA-P |
parkinsonism |
Nigrostriatal atrophy |
MSA-C |
Limb and gait ataxia |
Olivopontocerebellar atrophy |
Mixed subtype |
Both motor features |
Combined atrophy |
MSA-autonomic failure |
No motor features |
Central autonomic brain regions |
MSA-dementia |
No motor features |
Frontotemporal plus combined atrophy |
Reviewer comment on Page 3: "However, the density of OPCs appears to be highly correlated with the number of GCIs, suggesting a causative association between the presence of GCIs and the capacity to generate OPCs" --> How does density alone suggest causative association between presence of GCIs and capacity to generate OPCs? Can the authors elaborate a bit more?
Authors' note to Reviewer 2: we agree with the reviewer and have removed “causative” from this sentence.
Reviewer comment on" ..... This perhaps also explains the biological preference of forming GCIs in MSA rather than neuronal inclusions as observed in PD and DLB if more immature OPCs are indeed the origin of pathologic seeds of a-synuclein" --> the presence of more or less OPCs does not explain a preference for forming GCIs in MSA instead of neuronal inclusions. Recently, Azevedo and colleagues demonstrated that important cellular pathways and network dysfunction occur in MSA iPSC-derived OPCs which clearly suggests cell autonomous dysfunction as a starting point for the build-up of aggregates. Interestingly, the cells adopted an antigen presenting phenotype due to the aggregates formation and were delayed in their maturation, in line with work from Ettle, Winkler and colleagues.
Authors' note to Reviewer 2: In this paragraph, we discuss whether there is a rationale that human OPC can be a potential original of synuclein pathology, since the proliferative potential of these precursor cells in the adult and aged human brain may decline with aging. Having considered the reviewer’s suggestion, we now rearranged the order of sentences to ensure the logic flows smoother for this paragraph. We also noticed the same interesting findings of the two references recommended by the Reviewer, as shown in our original references 119 (Azevedo et al) and 31 (Ettle et al). In agreement with Reviewer’s opinion, we cited reference 31 in the statement sentence “Indeed, the impairment of OPC maturation has been suggested as a significant underlying mechanism contributing to the pathogenesis of MSA [28, 31]” on the same page. We also cited reference 119 in the later Section especially focusing on OPCs and iPSC models on page 10.
Reviewer comment on Page 6: figure 1 legend. please correct typo: OLGs should be replaced by OLCs
Authors' note to Reviewer 2: thanks for the reviewer pointing out this error and we have corrected this Figure legend.
Reviewer comment on Line 328: change inhibitor for inhibition
Authors' note to Reviewer 2: thanks for the reviewer pointing out this error and we have corrected it.
Reviewer comment on Line 413: remove extra space
Authors' note to Reviewer 2: thanks for reviewer pointing out this error and we have corrected it.
Reviewer comment on Line 414-415: This tripotential basal multipotent progenitor cell may provide further insights into the anosmia associated with a-synucleinopathies. --> please explain this sentence. what is the relation between anosmia and the formation of OPCs and tripotent stem cells? alternatively, please delete the whole paragraph except the first sentence.
Authors' note to Reviewer 2: we have deleted this sentence as the reviewer suggested and combined the remaining sentences in this paragraph with the next paragraph.
Reviewer 3 Report
I congratulate the Authors for their extensive and didactic review of the litterature on OPCs & oligodendrocytes in relation with MSA, the formation of GCIs and the processes of myelination & neurodegeneration. I did learn a lot from their MS. However, to update their comprehensive effort, the Authors might be willing to have a look to 3 very recent articles they might have missed and that perhaps will expand their overall reflexion and conclusions
https://doi.org/10.1038/s41593-022-01023-7
https://doi.org/10.1073/pnas.2202580119
https://doi.org/10.3390/biom13020269
Author Response
Reviewer comment "Authors might be willing to have a look to 3 very recent articles they might have missed and that perhaps will expand their overall reflexion and conclusions
https://doi.org/10.1038/s41593-022-01023-7
https://doi.org/10.1073/pnas.2202580119
https://doi.org/10.3390/biom13020269"
Authors' note to Reviewer 3: thanks for reviewer driving our attention to these new publications. We have included these articles in our Review as references [56, 57] in section 4.2. and reference 97 in section 7.